# Evaluation of the Immunological Efficacy of an LNP-mRNA Vaccine Prepared from Varicella Zoster Virus Glycoprotein gE with a Double-Mutated Carboxyl Terminus in Different Untranslated Regions in Mice

**DOI:** 10.3390/vaccines11091475

**Published:** 2023-09-11

**Authors:** Yunfei Wang, Han Cao, Kangyang Lin, Jingping Hu, Ning Luan, Cunbao Liu

**Affiliations:** Institute of Medical Biology, Chinese Academy of Medical Sciences and Peking Union Medical College, Kunming 650118, China; wangyf@imbcams.com.cn (Y.W.); caohan@imbcams.com.cn (H.C.); linky6679@163.com (K.L.); hujingping430@126.com (J.H.)

**Keywords:** VZV, mRNA vaccine, CMI, UTR

## Abstract

Cell-mediated immunity (CMI) plays a key role in the effectiveness of varicella zoster virus (VZV) vaccines, and mRNA vaccines have an innate advantage in inducing CMI. Glycoprotein E (gE) has been used widely as an antigen for VZV vaccines, and carboxyl-terminal mutations of gE are associated with VZV titer and infectivity. In addition, the untranslated regions (UTRs) of mRNA affect the stability and half-life of mRNA in the cell and are crucial for protein expression and antigenic translational efficiency. In this study, three UTRs were designed and connected to the nucleic acid sequence of gE-M, which is double mutated in the extracellular region of gE. Then, mRNA with different nucleic acids was encapsulated in lipid nanoparticles (LNPs), forming three LNP-mRNA VZV vaccines, named gE-M-Z, gE-M-M, and gE-M-P. The immune response elicited by these vaccines in mice was evaluated at intervals of 4 weeks, and the mice were sacrificed 2 weeks after the final immunization. In the results, the gE-M-P group, which retains the nucleic acid sequence of gE-M and is connected to Pfizer/BioNTech’s BNT162b2 UTRs, induced the strongest humoral immune response and CMI. Because CMI is crucial for protection against VZV and for the design of VZV vaccines, this study provides a feasible strategy for improving the effectiveness and economy of future VZV vaccines.

## 1. Introduction

Varicella zoster virus (VZV) can induce two kinds of diseases [1]. One is varicella, which occurs at the time of VZV initial infection and is common in children. In fact, because the virus is so contagious, nearly every adult has been infected with VZV before reaching adulthood and is a possible carrier [2,3]. VZV can hide in the ganglia after infection and be reactivated when the body’s immune system weakens or declines, which causes another disease called herpes zoster (HZ) and leads to symptoms of shingles such as fever, fatigue, local lymphatic pain, burning of the skin, neuralgia, and banded erythema along the path of the nerve. Accordingly, VZV vaccines, especially those that can efficiently induce cell-mediated immunity (CMI), play a key role in controlling latent infection of the virus in the body [4,5,6,7,8,9].

VZV is a herpes A virus with a 125 kbp genome encoding at least 70 unique open reading frames (ORFs). Among them, glycoprotein E (gE) is a type I essential membrane protein with 623 amino acids encoded by ORF68, which is highly expressed and important for VZV replication [10,11,12,13]. gE can freely move between the endoplasmic reticulum (ER), trans-Golgi network (TGN), and endosome, and thus could be an ideal antigen for VZV vaccines. In studies that use gE as an antigen, mRNA vaccines show immunogenicity comparable to that of AS01B-adjuvanted subunit vaccines, including gE-specific IgG titers and T-cell responses. In addition, mRNA vaccines can be manufactured without protein purification, allowing a rapid product development cycle, and the strong innate immune activation ability of mRNA can induce key protective acquired immune responses, including CMI, more effectively than traditional vaccines [14,15,16,17], making mRNA vaccines an excellent candidate for an economical and safer VZV vaccine [18,19].

Other researchers have shown that a carboxyl-terminal mutation of gE (gE-M) can improve the virus titer, the infectious ability of the virus at the cell surface, and the transportation and localization abilities of the virus. Additionally, the extracellular region of gE (gE-E) has been used as an antigen by the subunit vaccine Shingrix [20,21,22], mainly to improve the problems of protein purification. Therefore, we compared the efficacy between three mRNA VZV vaccines based on different coding sequences of gE, including full-length gE, gE-E, and gE-M, among which gE-M obtained the highest immune stimulation effect [19]. Accordingly, the nucleoid sequence of gE-M was used in our subsequent studies on VZV mRNA vaccines.

In addition, optimization of the mRNA vaccine antigen nucleic acid sequence, including the 5′ cap, 5′-UTR, 3′-UTR, and Poly(A) tail, has significant effects on antigen transcription, translation efficiency, spatial structure of antigen formation and the final immune effect [23]. BNT162b2 and mRNA-1273, the two mRNA vaccines for SARS-CoV-2 generated by Pfizer/BioNTech and Moderna, respectively, share the same amino acid sequence yet differ in many other ways, such as the ionizable lipid carriers, mRNA dosage, and 5′-UTR and 3′-UTR. Notably, although mRNA-1273 contains approximately three times as much mRNA as BNT162b2 (100 µg versus 30 µg), the efficacy of both vaccines against wild-type-induced COVID-19 was similar (95% vs. 94%). In this study, we referenced the sequences of the 5′-UTR, 3′-UTR, and poly(A) tails from the Pfizer/BioNTech (P) and Moderna (M) SARS-CoV-2 vaccines, as discussed above, and synthesized LNP-mRNA VZV vaccines named gE-M-P and gE-M-M, respectively. Additionally, a modified Zika virus (ZIKV) mRNA vaccine was designed by J M. Richner et al. and was shown to protect against ZIKV infection and confer sterilizing immunity [24]. The sequence of UTRs (including the 5′-UTR, 3′-UTR, and poly(A) tail) is also referenced in this study and named gE-M-Z. In conclusion, using Shingrix as a positive control, we showed that three LNP-mRNA VZV vaccines with different UTRs induced higher humoral immunity and CMI responses than Shingrix, showing that the LNP-mRNA VZV vaccines in this study could be promising and economical zoster vaccine candidates.

## 2. Materials and Methods

### 2.1. Vaccine Preparation

DNA sequences encoding the extracellular domain of gE (gE-E, 1-538 aa) with a C-terminal double mutant (mutant Y569A with the original motif AYRV, which targets gE to TGN, and mutants S593A, S595A, T596A and T598A with the original motif SSTT, which targets gE to the TGN or plasma membrane) were codon-optimized and synthesized by Sangon Biotech Co., Ltd., Shanghai, China. On the backbone of gE-M, three combinations of 5′-UTR, 3′-UTR, and poly(A) tail from ZIKV mRNA vaccine (Z), Pfizer/BioNTech BNT162b2 mRNA vaccine (P), and Moderna’s mRNA-1273 vaccine (M) were constructed, named gE-M-Z, gE-M-P and gE-M-M, respectively [15,18,25,26]. The structural diagram and regional differences between groups are shown in Figure 1.

mRNA was synthesized in vitro using T7 polymerase-mediated DNA-dependent RNA transcription by HyperScribe^TM^ All-In-One mRNA Synthesis Kit II Plus 1. In addition, the Monarch^®^ RNA Cleanup Kit (NEW ENGLAND BioLabs Inc., Ipswich, MA, USA) was used for purification of post-transcriptional mRNA, resulting in an mRNA quantity of 200 μg per sample. LNP-mRNA vaccines were prepared using the modified encapsulation procedure as previously described [19]. Briefly, lipids (from AVT Medical Technologies LTD, Tel Aviv, Israel) were dissolved in ethanol at molar ratios of 50:10:38.5:1.5 (MC3:DSPC:cholesterol:DMG-PEG2000). The lipid mixture was combined with 100 mM citrate buffer (pH 4.0) containing mRNA at a ratio of 3:1 (water: ethanol) using a microfluidic mixer (Precision Nanosystems, Vancouver, BC, USA). Then, vaccines were dialyzed using 40× volume PBS dialysis preparation, centrifuged at 3000 rpm, and subjected to ultrafiltration concentration using centrifugal filtration tubes (Millipore, Dublin, Ireland). Finally, vaccines were filtered through a 0.22 µm syringe filter for sterilization and stored at −80 °C until use. Particle size and polydispersity index (PDI) were measured using Malvern Panalytical (Malvern, UK). The mRNA encapsulation was detected by 1% denatured agarose, and the mRNA nucleic acid load was measured by the QuantiT^TM^ RiboGreen^®^ RNA Reagent and Kit (Thermo Fisher, Pleasanton, CA, USA). The encapsulation rate was calculated by comparing the detected nucleic acid load with the initial amount of nucleic acid added to the citric acid buffer.

### 2.2. Animal Studies

Six-week-old female specific pathogen-free (SPF) C57BL/6N mice (15–18 g) were purchased from Vital River Laboratory Animal Technology Ltd. (Beijing, China), randomly divided into groups of 6 mice each (*n* = 6), maintained under SPF conditions and housed with free access to food and water at the Central Animal Services of the Institute of Medical Biology, Chinese Academy of Medical Sciences (IMB, CAMS). The GSK vaccine group (purchased from GSK, MD, USA) was used as a positive control. PBS was used as a blank control group. One missing data point in group gE-M-M resulted from one mouse dying for unknown reasons before the final sacrifice. The mice were immunized intramuscularly (i.m.) in the thigh muscle twice with 50 µL immunogen at 4-week intervals. After anesthetization by intraperitoneal (i.p.) injection of tribromoethanol, blood samples (via cardiac puncture) and spleens were collected 2 weeks after the final immunization for further analysis.

### 2.3. Detection of Antibody Titers

After whole blood clotting at 4 °C overnight, immunized sera were collected after centrifugation at 3000 rpm for 20 min. gE (2 µg/mL, supplied by AtaGenix Laboratory Co., Ltd., Wuhan, China) was precoated on 96-well plates (Corning, Corning, NY, USA) at 4 °C overnight. After blocking with 5% (*w*/*v*) skim milk at 37 °C for 1 h, plates were incubated with 2 serial dilutions of mouse sera at 37 °C for 1 h. Bound antibodies were detected with goat anti-mouse IgG-horseradish peroxidase (HRP) conjugate (1:10,000, Bio-Rad, Hercules, CA, USA) as a secondary antibody. Ten min after the addition of the substrate 3,3′,5,5′–tetramethylbenzidine (TMB, BD, USA), 2 mol/L sulfuric acid was added to terminate the reaction. The absorbance at 450 nm was detected with a spectrophotometer (BioTek Instruments, Inc., Winooski, VT, USA). IgG titers were defined as end-point dilutions showing cutoff signals above OD_450_ = 0.15, and IgG titers lower than 200 were defined as 200 for calculations.

### 2.4. Enzyme-Linked Immunospot Assay (ELISPOT) and Enzyme-Linked Immunosorbent Assay (ELISA) of Splenocytes

Mouse spleens were dispersed with a 70 µm cell strainer (BD, USA). After red cell lysis by ammonium chloride potassium lysis buffer at room temperature (RT) for 5 min, splenocytes were resuspended in serum-free medium for universal ELISPOT (DAKEWE, Beijing, China) at a final concentration of 3 × 10^5^ cells/well in a 96-well plate (Millipore, Burlington, MA, USA). Then, an ELISPOT assay kit (BD, USA) was used according to the manufacturer’s protocol. gE at a final concentration of 20 µg/mL was added to splenocytes and incubated overnight. Spots were counted with an ELISPOT reader system (Autoimmun Diagnostika GmbH, Straßberg, Germany) after immunoimaging.

For the ELISA, splenocytes were resuspended in Roswell Park Memorial Institute (RPMI) 1640 medium with 10% *v*/*v* fetal bovine serum (FBS) (both from Biological Industries, Beit HaEmek, Israel) and 1% penicillin–streptomycin (Thermo Fisher, USA) at a final concentration of 1 × 10^7^ cells/mL. Then, 100 µL of cells was added to each well of a 96-well plate (Corning, USA). gE at a final concentration of 10 µg/mL was added to each well, and the same volume of PMA+ ionomycin (DAKEWE, China) was used as a positive control. After incubation for 24 h at 37 °C in 5% CO_2_, supernatants were collected, and cytokine levels were detected by ELISA. Briefly, unconjugated IL-2 (3 µg/mL) and IFN-γ (4 µg/mL) antibodies (Invitrogen, Waltham, MA, USA) dissolved in PBS were coated onto 96-well plates at 4 °C overnight. After blocking the plate with 1% (*w*/*v*) BSA at 37 °C for 1 h, 50 µL/well cell supernatant was added to each well and incubated for 3 h at RT. Biotin-conjugated antibodies against IL-2 and IFN-γ (2 µg/mL, Invitrogen) dissolved in 1% BSA were incubated for 1 h, and HRP-conjugated streptavidin (1 µg/mL, Biolegend, San Diego, CA, USA) was incubated for 30 min. Finally, after ten min at RT after addition of the TMB substrate, 2 mol/L sulfuric acid was added to terminate the reaction.

### 2.5. Flow Cytometry

Splenocytes were resuspended and seeded in plates (1 × 10^6^ cells/mL), gE at a final concentration of 10 µg/mL was added, and the two were co-incubated for another 2 h. Brefeldin A (5 µg/mL) was added and incubated overnight to block cytokine release. Cells were collected and subjected to viability staining using Zombie NIR™ dye to discriminate between live and dead cells. After the addition of 5 µg/mL CD16/CD32 antibodies to block the nonspecific binding of Fc receptors by incubation at 4 °C for 10 min, PerCP/Cyanine 5.5-tagged anti-mouse CD4 and FITC-tagged anti-mouse CD8 were added and incubated at 4 °C for another 30 min. After fixation with 4% formaldehyde at room temperature for 20 min, cells were washed with permeabilization wash buffer. PE-tagged anti-mouse IFN-γ and APC-tagged anti-mouse IL-2 antibodies were added and incubated. More than 20,000 CD4^+^ or CD8^+^ T-cell events were analyzed with a CytoFLEX flow cytometer (Beckman, Indianapolis, IN, USA) and FlowJo software (Version 10.6.2, BD, NJ, USA).

### 2.6. Statistical Analysis

Data were analyzed with one-way analysis of variance (ANOVA) followed by Dunnett’s multiple comparisons test, with the Shingrix group as the control. GraphPad Prism 8.0 (GraphPad Software Inc., La Jolla, CA, USA) was used for statistical analyses.

## 3. Results

### 3.1. LNPs Efficiently Encapsulated mRNA Antigens with Uniform Particle Sizes

The diameters of LNPs encapsulating gE-M-Z, gE-M-M, and gE-M-P mRNA were 93.30 nm, 83.03 nm, and 84.52 nm, respectively (Figure 2A). All of the polydispersity indexes (PDIs), which are measures of the heterogeneity of a sample based on size, were lower than 0.15 (gE-M-Z: 0.149, gE-M-M: 0.119, and gE-M-P: 0.117, Figure 2B), indicating good uniformity of the nanoparticles. Moreover, the encapsulation efficiencies were higher than 80% (87.8% for gE-M-Z, 90.5% for gE-M-M, and 84.2% for gE-M-P, Figure 2C). All the encapsulated LNP-mRNA vaccines showed good integrity on denatured agarose gel as bands at approximately 2000 bases, with gE-M-P being slightly higher (Figure 2D). After lysis by 1% Triton overnight at 4 °C, agarose gel electrophoresis showed that the mRNA in the LNP vaccine degraded only slightly, indicating good integrity (Figure 2D). All results in Figure 1 show that LNPs effectively encapsulated the mRNA vaccine and formed uniform particle sizes, stable structures, and completely wrapped LNP-mRNA nanoparticles.

### 3.2. mRNA Vaccines with Different UTRs Affect Humoral Immune Responses

In the detection of gE-specific IgG antibody titers, the gE-M-M group had the highest average (870,400), followed by the gE-M-Z group (725,333), and the gE-M-P group had the lowest (469,333). The Shingrix group vaccine had a higher gE-specific IgG antibody titer (1,365,330) than all other groups (Figure 3), approximately 1.56 times that of the gE-M-M group, 1.88 times that of the gE-M-Z group and almost 3 times that of the gE-M-P group. There was no significant difference between the gE-M-Z, gE-M-P, and Shingrix groups (*p* = 0.07 and *p* = 0.22, respectively), and there was an extremely significant difference between the gE-M-P and Shingrix groups (*p* = 0.009, **). Meanwhile, the antibody titers of all the groups were greater than 450,000, which indicates that although the LNP-mRNA vaccines were weaker than the Shingrix group, they also stimulated strong humoral immunity.

### 3.3. Among LNP-mRNA Vaccines with Different UTRs, gE-M-P Mediates the Highest CMI

Two cytokines, IL-2 (Figure 4A) and IFN-γ (Figure 4C), in the supernatant of splenocytes were tested by ELISA. The gE-M-P group had the highest induced cytokine activity, while there was no significant difference between the gE-M-P group and the Shingrix group. Specifically, the average values of IL-2/IFN-γ were 0.445/0.321 ng/mL in the gE-M-Z group, 0.345/0.424 ng/mL in the gE-M-M group, and 0.509/0.894 ng/mL in the gE-M-P group, which were all higher than those of the Shingrix group (0.150/0.147 ng/mL).

Similar trends were seen in the ELISPOT results. In terms of the average value of IL-2 spots (Figure 4B), the gE-M-P group showed the highest mean value (319.8), followed by the gE-M-Z group (181.7) and the gE-M-M group (155). The mean value of all three LNP-mRNA vaccine groups was higher than that of the Shingrix group (88.5). For IFN-γ spots, all three LNP-mRNA vaccines achieved greater effects than the Shingrix group, and the average values were 273, 261.2, 494, and 95.5 in the gE-M-Z group, gE-M-M group, gE-M-P, and Shingrix group, respectively (Figure 4D). There was a significant difference between the gE-M-P group and the Shingrix group in both IL-2 secretion (*p* = 0.01, *) and IFN-γ secretion (*p* = 0.03, *) for ELISPOT analysis. To directly observe the secretion levels of cytokines, blots of gE-specific cytokine secretion detected by ELISPOT are shown in Figure 4E. It can be seen that the gE-M-P group mediates the highest CMI among all groups.

### 3.4. Compared to Shingrix, the LNP-mRNA Vaccine Groups Showed Slightly Increased CD4^+^ and CD8^+^ T-Cells

Unlike foreign protein antigens that may pass through a cross-presentation pathway with limited efficiency, LNP-mRNA vaccines elicit the translation of proteins in the cytoplasm, which can then be processed into peptides and presented to major histocompatibility complexes type I (MHC-I) in the same way as heterologous antigens produced by viral infection, resulting in the innate ability to stimulate strong humoral and cellular immunity, among which CD4^+^ T-cells and CD8^+^ T-cells are important detection indicators. Flow cytometry was used to detect the positive rates of IL-2/IFN-γ-producing CD4^+^ and CD8^+^ T-cells in mouse splenocytes of different groups. Regarding CD4^+^ T-cells, the LNP-mRNA vaccine groups showed slightly higher numbers than the Shingrix group, but there was no significant difference by statistical analysis (Figure 5A,B). In cytokine-producing CD8^+^ T-cells, the LNP-mRNA vaccine groups achieved similar levels to the Shingrix group, among which the gE-M-P group showed slightly higher levels than the other groups with no significant difference shown by statistical analysis (Figure 5C,D). In conclusion, we can infer that gE-M-P was the best of the LNP-mRNA VZV vaccines for inducing efficient CMI.

## 4. Discussion

Approximately one-third of the population will be affected by zoster in their lifetime, and mental stress, economic loss, and life threats are caused by the onset of herpes zoster. Vaccination is the only way to prevent VZV infection [2]. There are three VZV vaccines currently on the market, two of which are live-attenuated vaccines (Zostavax produced by Merck in 2005 and Ganwei produced by Changchun BCHT Biotechnology Co. in 2023), and another which is a subunit vaccine (Shingrix produced by GSK in 2017). Based on real-world protection data, Shingrix induces a higher level of protection than Zostavax and became the most marketable VZV vaccine. However, the production capacity of Shingrix is extremely limited by its AS01b adjuvant system, because QS21, a key component of AS01b, is a natural product from the bark of Quillaja Saponaria and is hard and expensive to obtain and purify.

With the emergence of the COVID-19 epidemic, the development of mRNA vaccines has gained unprecedented enthusiasm owing to its simple production process, unrestricted access to raw materials, and effective induction of CMI [27]. Reports have demonstrated that the severity of HZ and its complications increases with age. Agedness would drive VZV-specific CMI to a low threshold that would be unable to suppress viral reactivation [2]. As CMI, but not humoral immunity, plays an important role in the effectiveness of VZV vaccines and the prevention of latent viral infection [6,28], mRNA vaccines are perfect candidates for the development of VZV vaccines. An mRNA form of the VZV vaccine has been tested in nonhuman primates and has shown comparable efficacy to that induced by VZV gE- and AS01B-adjuvanted subunit vaccines [18]. Two VZV mRNA vaccines entered Phase I and Phase I/II trials in January 2023 and on 10 February 2023, by Moderna and by Pfizer and BioNTech, respectively. Our study also showed that mRNA-based VZV vaccines could induce humoral (Figure 3) and cellular immunity (Figure 4) comparable to that of Shingrix, and the economic efficiency of VZV mRNA vaccines has been fully confirmed.

Nucleic acid sequences, including both the coding sequence that encodes protein antigens and the untranslated regions of the mRNA, can be designed and implemented quickly for new mRNA vaccines and are important factors for the efficiency of mRNA translation. In a previous study, we examined the effect of VZV gE carboxyl-terminal mutation on mRNA vaccine efficacy, and gE-M with a C-terminal double mutation (mutation Y569A with the original motif AYRV, which targets gE to TGN, and mutations S593A, S595A, T596A, and T598A with the original motif SSTT, which targets gE to the TGN or plasma membrane) was chosen as the ORF to be used in this study due to its advantages in all of the indicators tested [19]. In addition to the ORF, other untranslated regions of the mRNA sequence, such as the 5′-UTR, 3′-UTR, and poly(A) tail, are also critical for the efficiency of protein translation and expression. Accordingly, we designed three VZV gE-M-UTR constructs: gE-M-Z with ZIKV vaccine UTRs, gE-M-P optimized with Pfizer’s mRNA BNT162b2 UTR, and gE-M-M optimized with Moderna’s mRNA-1273 UTR (Figure 1). When encapsulated in LNPs, all three mRNA vaccines showed uniform characteristics, including diameter, PDI, and encapsulation efficiency (Figure 2).

After humoral immunity was evaluated by gE-specific IgG antibody titer and the cellular immune response was measured by ELISPOT, ELISA, and flow cytometry, our study showed that gE-M-P is the best mRNA vaccine among those three candidates. Although gE-M-P induced three-fold lower IgG titers than Shingrix, the ability of gE-M-P to induce CMI was significantly higher than that of the other groups. In the prevention of zoster, higher CMI has been demonstrated to be more important than IgG titers. As Shingrix requires two doses to achieve a full immunization schedule, we speculated that two doses of mRNA immunization would result in a much higher CMI and higher protection rate than Shingrix, which needs to be studied further.

## 5. Conclusions

In conclusion, the vaccine prepared in LNP-mRNA form with gE-M as the ORF has great innate advantages in the prevention of herpes zoster virus infection and recurrence. Since cellular immunity plays an extremely important role in preventing VZV latent infection and reactivation, the LNP-mRNA vaccine was a good choice for improving both humoral and cellular immune responses. Our study demonstrated that the LNP-mRNA vaccine prepared with gE-M as the ORF, with the UTRs of Pfizer’s mRNA BNT162b2, is a potent, feasible, and economic candidate for the VZV vaccine.

## Figures and Tables

**Figure 1 vaccines-11-01475-f001:**
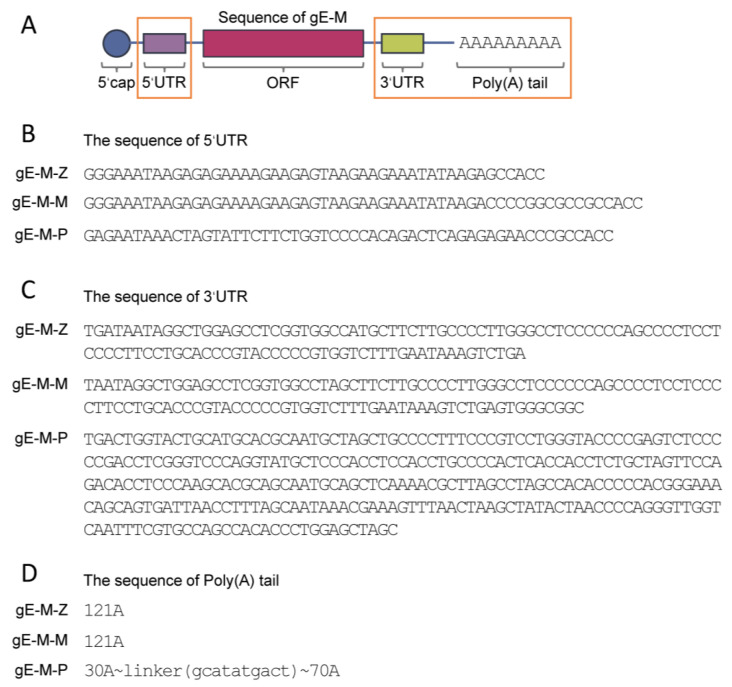
Diagram and sequence of LNP-mRNA VZV vaccines synthesized in this study. (**A**) Schematic diagram of the components of the synthesis of the nucleic acid sequences of mRNA. Red boxes indicate the altered parts in each group. (**B**–**D**) Sequence differences in the 5′-UTR, 3′-UTR, and poly (**D**) tail between groups, respectively. ORF: open reading frame; UTR: untranslated regions.

**Figure 2 vaccines-11-01475-f002:**
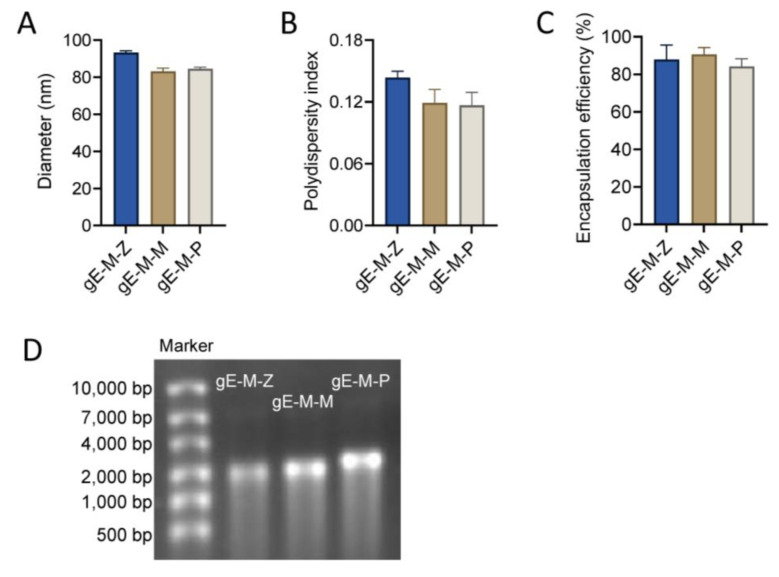
Characterization of LNP mRNA vaccines. The group labels were gE-M-Z and gE-M-M gE-M-P. (**A**) Diameters tested by size analyzer; (**B**) polydispersity index of LNPs; (**C**) mRNA encapsulation efficiency; (**D**) loaded mRNA detected with 1% denatured agarose gel electrophoresis.

**Figure 3 vaccines-11-01475-f003:**
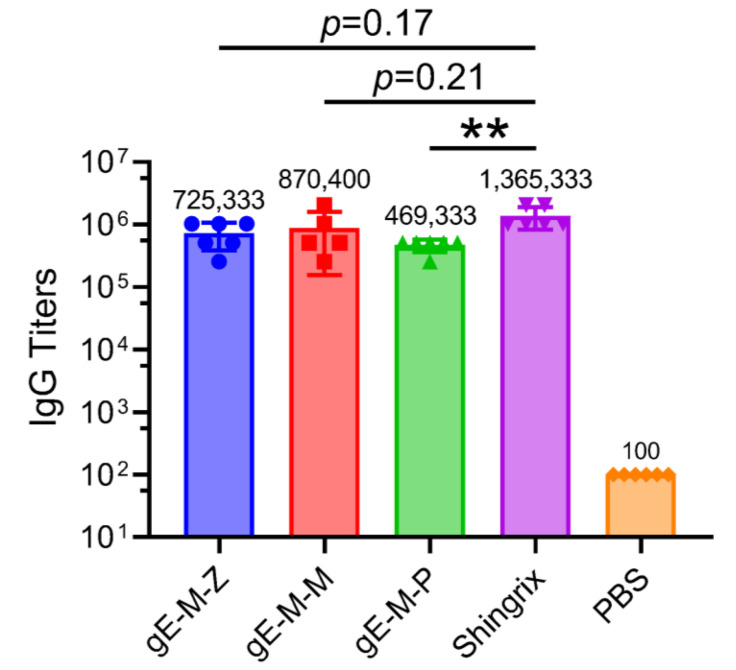
gE-specific IgG titers detected by enzyme-linked immunosorbent assay (ELISA). The mean value of each group was shown on the top of every bar. IgG titers were compared using one-way analysis of variance (ANOVA) followed by Kruskal–Wallis test, with the Shingrix group as a control. Each point represents an individual mouse, ** *p* < 0.01.

**Figure 4 vaccines-11-01475-f004:**
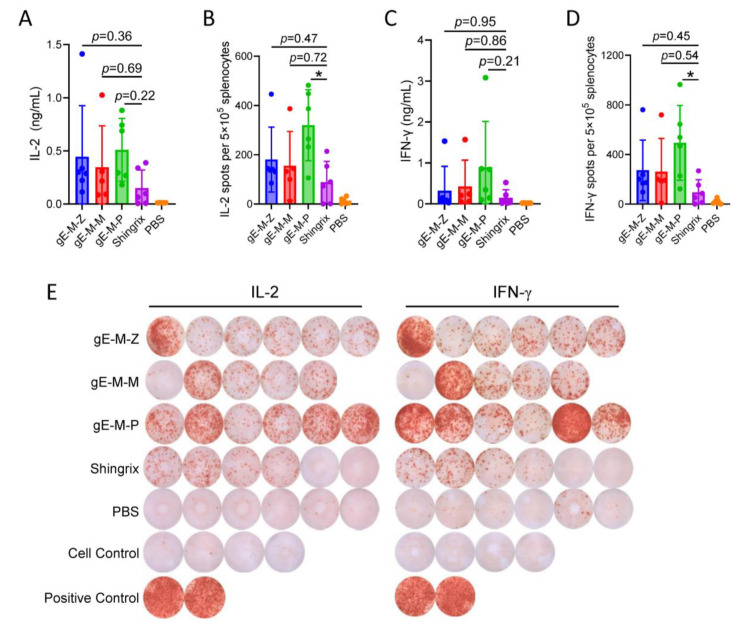
The secretion levels of IL-2 and IFN-γ in splenocytes were assayed by ELISA and ELISPOT. (**A**,**B**) The levels of IL-2 produced by splenocytes after gE stimulation; (**C**,**D**) the levels of IFN-γ produced by splenocytes after gE stimulation; (**E**) pictures of individual spots. Means were compared using one-way ANOVA followed by Dunnett’s multiple comparisons test, with the Shingrix group as a control. Points represent individual mice, * *p* < 0.05, *p* > 0.99, not significant.

**Figure 5 vaccines-11-01475-f005:**
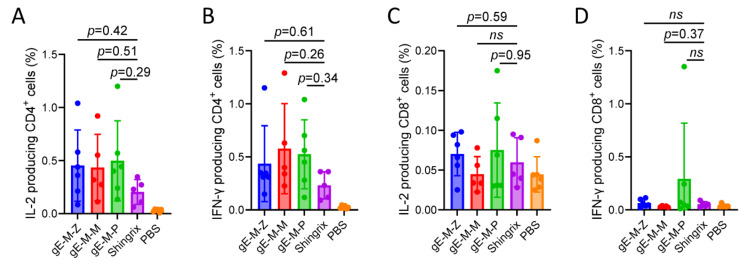
Flow cytometry assay for gE-specific IL-2- and IFN-γ-producing T-cells. (**A**,**B**) Proportion of IL-2- and IFN-γ- producing CD4^+^ T-cells among splenocytes after stimulation with gE. (**C**,**D**) Proportion of IL-2- and IFN-γ- producing CD8^+^ T-cells among splenocytes after stimulation with gE. Data were analyzed using one-way ANOVA followed by Dunnett’s multiple comparisons test, with the Shingrix group as a control. Not significant (ns) means *p* > 0.99.

## Data Availability

All data used during the study are available from the corresponding author by request.

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
