# Peer review of "Evaluation of the Immunological Efficacy of an LNP-mRNA Vaccine Prepared from Varicella Zoster Virus Glycoprotein gE with a Double-Mutated Carboxyl Terminus in Different Untranslated Regions in Mice"

_vaccines, 2023, doi:10.3390/vaccines11091475_

Round 1

Reviewer 1 Report

Finding a new vaccination strategy for Varicella is an important need and mRNA vaccines are novel and evolving. This manuscript has major flaws that need to be corrected prior to publication.

Justification for the use of only female mice needs to be made in the methods section. Using only one sex of mice is antiquated and limits important variability in data analysis.

In Figure 3, there was only one replicate of six mice. With the significance close on all samples, it seems that another experimental replicate would be necessary to draw conclusions on titer significance.

For you cytokine ELISA, milk block was used with a biotinylated antibody. Milk block should not be used with biotinylated antibodies. (Inhibition of the streptavidin-biotin interaction by milk

W L Hoffman et al. Anal Biochem. 1989 Sep.)

The large variation in the data with all three of the mRNA vaccinations paired with one experiment and only the low experimental n, it is my recommendation that these experiments are repeated once more with a full repeat. The current data makes drawing conclusions at all difficult.

For flow cytometry analysis, a live/dead exclusion marker was not utilized. This is especially important when considering a restimulation protocol where brefeldin A is utilized for an extensive amount of time. Without the use of a live/dead exclusion marker, the flow cytometry data is not acceptable. Additionally, essential steps were left out of the methods, including fixation prior to permeabizing for intracellular target staining. I assume the authors completed this step, but it needs to be included. A gating strategy included in supplemental data would also allow assessment of the flow cytometry to assure that the data was correctly analyzed. FMOs were not mentioned and these need to be used to gated intracellular targets and are considered standard practice. Without all of these changes, the flow cytometry data cannot be interpreted.

The discussion and conclusions are overstated about which vaccination is “the best.” There is intense variability and the majority of the data is not conclusive. 

Author Response

I'm honored to have your review. I have corrected the major flaws you mentioned in the revised manuscript, and answered your questions as bellows:

  1. With regard to the sex and numbers of experimental mice in this study, you pointed out that we only use female mice and six for one group may limit important variability in data analysis. Thanks for your valuable opinion, we have noticed the limit in our experiment design, and the importance of sex and numbers will be paid more attention in our future study. In this study, we reference our past researches, in which sex of experimental mice didn’t influence results much, and six mice in one group are matching the ‘3R principle’ in animal experiment.
  2. You mentioned that the significance close on all samples in figure 3 was caused by the number of mice. In fact, the IgG titers wouldn’t change a lot in orders of magnitude, and we lied more emphasis on cell immune response, but not humoral immune responses in this research, due to the cellular immune response is more important than humoral immune response for VZV vaccine development.
  3. Thank you for your mention that we realized that we made the mistake. In our study, 5% skim milk and 1% BSA were used for Elisa in IgG titers detection and cytokine detection, respectively. We mistake markup in cytokine detection part, and we changed it in line 189 and line 191 in the revised manuscript.
  4. Thanks for your kindly suggestion. As we explained above, six mice in one group are matching the ‘3R principle’ in animal experiment, and our past results also showed that six mice can get more reliable and stable data than four or five mice in one group, without losing accuracy.
  5. We did stain cells by using Zombie NIR™ dye to exclude live/dead cells, and fix cells prior to permeabilization for intracellular target staining, methods have been restated in line 198 and line 203, respectively. In general, FMOs are used in multicolor experiments, which will cause signals of certain channels difficult to distinguish. In our flow cytometry, channels are easy to distinguish by setting unstaining control and single dye controls, regulation of compensation and analyze of positive indicators can be easy accomplish by our gating methods.
  6. Thanks for your mention that we realized that our statement is not scientific enough. We changed it in line 360 in the revised manuscript.

Reviewer 2 Report

The article provides an in-depth description of the development of VZV vaccines, with a particular emphasis on the utilization of LNP-mRNA technology. The exploration of mRNA technology within the realm of vaccine development represents a relatively novel approach, especially concerning VZV. The design and testing of three new types of mRNA vaccines in the study mark significant innovative strides. A clear comparison has been made between Shingrix and the novel mRNA vaccines. One of the LNP-mRNA vaccines (gE-M-P) demonstrated the best performance in inducing cell-mediated immunity (CMI), even though its IgG levels were three-fold lower than those of the commercial vaccine Shingrix. The gE-M-P mRNA vaccine, despite inducing IgG antibody levels three-fold lower than Shingrix, exhibited significantly higher ability in inducing cellular immunity compared to other vaccine groups.

There is a need to explain and ascertain if other mRNA vaccines present a similar situation, and further elaboration in the discussion would be beneficial.

More evidence is needed to prove the vaccine's effectiveness or protection, especially considering large standard deviations (SD). An explanation regarding the statistical variations may be required to bolster the study's conclusions.

In terms of serological evidence, it is recommended to use internationally recognized standards (such as standard sera from convalescent patients) for methodological validation, or conduct real virus neutralization ability tests, which would lend further credence to the study's findings and enhance the overall understanding of the vaccine's protective capacity.

No comment

Author Response

Thank you for your professional comments. Please check our responses to your comments as follows:

  1. Thanks for your advice. In the designing of mRNA vaccines, 5'UTR directly affects the translation of its downstream open reading frame (ORF) and improves the stability of mRNA. 3'UTR will affect the stability of mRNA through its interaction with RNA binding protein, thus affecting the translation efficiency. The 5'UTR and 3'UTR structures of Pfizer-162b2 and Moderna mRNA-1273 are different. However, there is no related comparison study about whether different UTR sequence will directly lead the change of immunological induced by mRNA vaccines, in the basis of other sequence of mRNA vaccines are same.
  2. As we known, standard deviations (SD) could represent the individual data distributed around mean, hence we choose mean± SD to describe data of animal experiment in this study. We thought that large SD is considerable because the high individual variability between different mice. Indeed, we need more experiments, such as real VZV virus to attack immunized mice, to prove the vaccine's effectiveness or protection, which will be our future goal to enable further development of our lab-made mRNA VZV vaccines.
  3. Thanks for your thoughtful recommendation, your suggestion will made our results more reliable and get more internationally recognition. Real virus neutralization ability tests truly will enrich our results, but we pay more attention on the cellular immune response, such as CD4 and CD8 splenocyte, but not humoral immune response. Cellular immune response is more important than humoral immune response to provide protective capacity for an VZV vaccine.

Reviewer 3 Report

Estimated Authors,

I've read this basic research study with great interest. VZV is a very common and often underestimated pathogen, and the available vaccines are affected by substantial shortcomings - mainly, the high cost per dose.

The present study suggest that a tentative mRNA vaccine against antigen E could provide acceptable vaccination efficacy, allowing the design or more appropriate vaccination campaigns.

Unfortunately, the present study is affected by shortcomings that require some amendments before its eventual acceptance. However, I'm confident that Authors could provide the adjustments I'm suggesting in a very short timeframe.

More precisely, the most significant issue is based on the statistical analysis. According to the methods, Authors collected a total of 6 mice per group, and IgG levels / efficacy were then assessed by means of ANOVA with Dunnet's test. Even though Dunnet's test, by its design, could cope with the reduced number of samples, and despite the genetic analogies across the lab mice, a non-parametric test should be preferred (e.g. Kruskal Wallis test).

After the revision of the statistical analyses, Authors should double check the histograms in order to make them more readable. As only 6 specimens per group are included, please design the histogram in order to make the individual estimates more properly visible (for example: Fig. 5 A do D, you could make the colors of histogram more clear in order to make more evident the individual estimates).

Similarly, please be aware that, when providing electrophoresis gel assays, the quality of the images should be particularly high, and Figure 2D is (at least in my copy) of very low quality. Please provide a high resolution copy - at least as supplementary material.

Eventually, please discuss more extensively how the dichotomy between efficacy and concentration of antibodies could be explained.

Nothing to address.

Author Response

We appreciate very much for your professional comments. Minors has been revised as bellows:

  1. We have changed the statistical analysis to non-parametric test (Kruskal Wallis test) in Figure 3, revised figure has been attached in line 248. It seems like different means of ANOVA didn’t influence much on the result.
  2. To made histograms more readable, we changed the figures to the more contrast color, new figures have been attached in line 248, line 275, and line 303, respectively.
  3. We feel regret that the quality of electrophoresis gel was limited by our poor camera resolution. To make it clear, original gel image and image with contrast altered by photoshop were provided to the editorial office, we hope it meet publication requirements of Vaccines.
  4. In our experience, there is no necessary link between the protective efficiency of VZV vaccines and antibody titers.

Round 2

Reviewer 1 Report

All comments are addressed

Author Response

Thanks for your professional review comments.

Reviewer 2 Report

The author has provided a comprehensive and satisfactory response to all the queries raised, offering clear explanations for each point. I have reviewed these in detail, and I am satisfied with the revisions made to the manuscript.

Author Response

(The authors gave the same response as above.)
